 SHORT REPORT

# Selenocyanate derived Se-incorporation into the nitrogenase Fe protein cluster

Trixia M Buscagan[1,2]*, Jens T Kaiser[1], Douglas C Rees[1,2]*

[1]Division of Chemistry and Chemical Engineering, California Institute of Technology, Pasadena, United States; [2]Howard Hughes Medical Institute, California Institute of Technology, Pasadena, United States

**Abstract** The nitrogenase Fe protein mediates ATP-dependent electron transfer to the nitrogenase MoFe protein during nitrogen fixation, in addition to catalyzing MoFe protein-independent substrate ($CO_2$) reduction and facilitating MoFe protein metallocluster biosynthesis. The precise role(s) of the Fe protein $Fe_4S_4$ cluster in some of these processes remains ill-defined. Herein, we report crystallographic data demonstrating ATP-dependent chalcogenide exchange at the $Fe_4S_4$ cluster of the nitrogenase Fe protein when potassium selenocyanate is used as the selenium source, an unexpected result as the Fe protein cluster is not traditionally perceived as a site of substrate binding within nitrogenase. The observed chalcogenide exchange illustrates that this $Fe_4S_4$ cluster is capable of core substitution reactions under certain conditions, adding to the Fe protein's repertoire of unique properties.

## Editor's evaluation

This manuscript describes the unexpected observation of selenium exchange into an iron-sulfur cluster cofactor of a component of nitrogenase. The work sets the stage for future mechanistic study of this phenomenon. It also provides a roadmap for study of sulfide exchange in other classes of iron-sulfur cluster enzymes.

*For correspondence:
trixia.marie.b@gmail.com (TMB);
dcrees@caltech.edu (DCR)

**Competing interest:** The authors declare that no competing interests exist.

## Introduction

The nitrogenase Fe protein has multiple roles, with its most famous role being ATP-dependent electron transfer to the MoFe protein during $N_2$ fixation (***Figure 1***; ***Thorneley and Lowe, 1983***; ***Wolle et al., 1992***; ***Rutledge and Tezcan, 2020***). The Fe protein also catalyzes MoFe protein-independent $CO_2$-to-CO reduction (***Rebelein et al., 2017***), and participates in the biosynthesis of both the P-cluster and FeMo-cofactor (***Allen et al., 1993***; ***Burén et al., 2020***). Unlike most $Fe_4S_4$ clusters in metalloproteins which adopt two oxidation states, the Fe protein cluster can span three oxidation states (2+/1+/0) (***Watt and Reddy, 1994***; ***Angove et al., 1997***; ***Liu et al., 2014***). While both MgATP- and MgADP binding to the Fe protein result in lower reduction potentials of the $Fe_4S_4$ cluster relative to the nucleotide-free state (see ***Rutledge and Tezcan, 2020***), only the MgATP-bound state of the protein in the 1+ state is susceptible to rapid and complete iron chelation with bipyridine or bathophenanthroline (***Walker and Mortenson, 1974***; ***Ljones and Burris, 1978***; ***Hausinger and Howard, 1983***; ***Anderson and Howard, 1984***). In the absence of nucleotide, iron chelation is slow, while MgADP inhibits chelation. Furthermore, the 2+ oxidized form of the $Fe_4S_4$ cluster undergoes ATP-dependent Fe chelation, yielding an intact $Fe_2S_2$ cluster (***Anderson and Howard, 1984***). The origins of these unusual properties of the Fe protein cluster are not well understood, but may reflect the solvent accessibility of the cluster and its positioning at the dimer interface.

**eLife digest** Many of the molecules that form the building blocks of life contain nitrogen. This element makes up most of the gas in the atmosphere, but in this form, it does not easily react, and most organisms cannot incorporate atmospheric nitrogen into biological molecules. To get around this problem, some species of bacteria produce an enzyme complex called nitrogenase that can transform nitrogen from the air into ammonia. This process is called nitrogen fixation, and it converts nitrogen into a form that can be used to sustain life.

The nitrogenase complex is made up of two proteins: the MoFe protein, which contains the active site that binds nitrogen, turning it into ammonia; and the Fe protein, which drives the reaction. Besides the nitrogen fixation reaction, the Fe protein is involved in other biological processes, but it was not thought to bind directly to nitrogen, or to any of the other small molecules that the nitrogenase complex acts on. The Fe protein contains a cluster of iron and sulfur ions that is required to drive the nitrogen fixation reaction, but the role of this cluster in the other reactions performed by the Fe protein remains unclear.

To better understand the role of this iron sulfur cluster, Buscagan, Kaiser and Rees used X-ray crystallography, a technique that can determine the structure of molecules. This approach revealed for the first time that when nitrogenase reacts with a small molecule called selenocyanate, the selenium in this molecule can replace the sulfur ions of the iron sulfur cluster in the Fe protein. Buscagan, Kaiser and Rees also demonstrated that the Fe protein could still incorporate selenium ions in the absence of the MoFe protein, which has traditionally been thought to provide the site essential for transforming small molecules.

These results indicate that the iron sulfur cluster in the Fe protein may bind directly to small molecules that react with nitrogenase. In the future, these findings could lead to the development of new molecules that artificially produce ammonia from nitrogen, an important process for fertilizer manufacturing. In addition, the iron sulfur cluster found in the Fe protein is also present in many other proteins, so Buscagan, Kaiser and Rees' experiments may shed light on the factors that control other biological reactions.

Our group has reported a crystallographic approach for quantifying Se-incorporation into the active site FeMo-cofactor of the MoFe protein (*Spatzal et al., 2015*). Key to this study was potassium selenocyanate (KSeCN), which like thiocyanate, is an alternative substrate for nitrogenase (*Rasche and Seefeldt, 1997*; *Spatzal et al., 2015*). Within nitrogenase, the FeMo-cofactor is traditionally perceived as the site of $N_2$ (and other substrate) binding. The observation that Se-incorporation occurred at the FeMo-cofactor under KSeCN turnover, but not at the P-cluster, supported this paradigm. Herein, using these conditions, we report a novel cluster conversion at the Fe protein in which the sulfide ligands of the $Fe_4S_4$ cluster exchange with 'Se' from KSeCN to yield an intact $Fe_4X_4$ cluster (X = Se, S) with Se-incorporation at all chalcogenide sites. This result was unexpected as the Fe protein cluster is not traditionally considered a substrate-binding site. While the generation of $Fe_4Se_4$-containing Fe proteins using apoproteins (proteins deficient in the native $Fe_4S_4$ cluster) and a (1) selenium source, iron source, and reductant or (2) with synthetic clusters has been reported (*Hallenbeck et al., 2009*; *Solomon et al., 2022*), the work described herein details a reaction distinct from reconstitution; namely, we report an exchange reaction under KSeCN turnover using native $Fe_4S_4$-containing Fe protein.

## Results

We initially observed Se-incorporation into the Fe protein cluster using our group's previously reported KSeCN turnover conditions, which include KSeCN as the selenium source, dithionite as the reductant, and an ATP regenerating system (*Mustafa and Mortenson, 1967*; *Spatzal et al., 2015*). Crystallization of the nitrogenase proteins from the concentrated reaction mixture was achieved by selecting conditions that favor either MoFe protein or Fe protein crystals (*Wenke et al., 2019a*; *Wenke et al., 2019b*). The crystal structure at 1.51 Å resolution of the Se-incorporated Fe protein isolated from this reaction mixture is shown in *Figure 2*. The crystal form is isomorphous to the previously reported

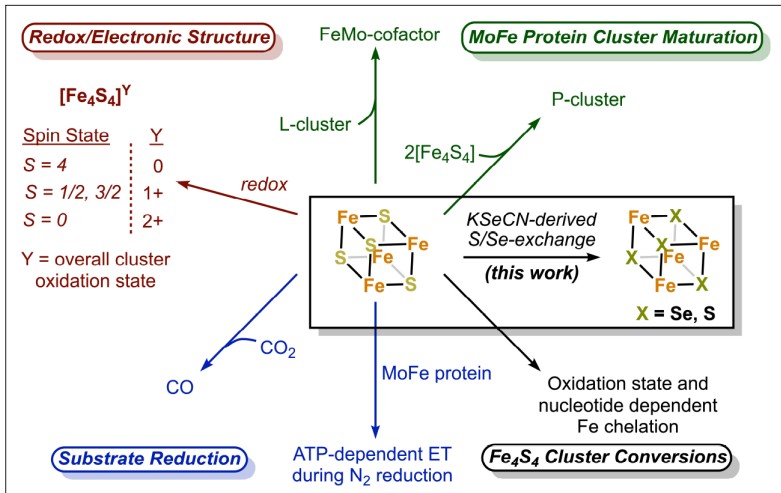

**Figure 1.** The nitrogenase Fe protein contains a Fe₄S₄ cluster with unique properties and participates in multiple reactions.

MgADP-bound state of the Fe protein (*Wenke et al., 2019b*), with the Fe protein molecular twofold axis coincident with a crystallographic twofold axis so that the asymmetric unit contains one subunit and half the cluster. The unique Fe1 and Fe2 sites are coordinated to Cys 97A and Cys 132A, respectively, while the unique chalcogenide sites 3 and 4 are buried and surface exposed, respectively. The locations of the Se ions within the protein structure were identified by collecting two sets of anomalous diffraction data: one above (12,668 eV) and one below (12,643 eV) the Se K-edge. Well-defined density was observed at both chalcogenide positions of the $Fe_4S_4$ cluster in the double difference anomalous Fourier map ($\Delta anom_{12,668\ eV} - \Delta anom_{12,643\ eV}$). Modeling the cluster exclusively as either the $Fe_4S_4$ or $Fe_4Se_4$ form resulted in substantial positive or negative difference density in the corresponding $F_{obs} - F_{calc}$ difference Fourier maps, respectively (*Figure 2—figure supplement 1*). Likewise, *B*-factors with lower or higher values at the core chalcogenide positions, relative to the iron cluster positions, were observed when the cluster was modeled exclusively as the all-sulfide vs. all-selenide form, suggesting an under- vs. over-modeling of electron density, respectively (*Supplementary file 1*). By fixing the chalcogenide *B*-factor values to a value similar to that of the Fe ions, satisfactory mixed cluster models were obtained (see Methods for refinement details, *Supplementary file 2*, and *Figure 2—figure supplement 2*). The Se occupancies at the X3 and X4 positions are shown in *Table 1*, entry 1, with the buried X3 position exhibiting a greater extent of Se-incorporation relative to the surface exposed X4 position.

To discern the essential components for Se-incorporation at the Fe protein cluster, control reactions were performed and the resultant protein crystallized and subjected to X-ray diffraction (XRD). To determine whether the MoFe protein was required for Se-incorporation at the Fe protein cluster, the MoFe protein was omitted from the reaction (*Table 1*, entry 2). Se-incorporation at the Fe protein cluster occurred in the absence of the MoFe protein as observed in the $\Delta anom_{12,668\ eV} - \Delta anom_{12,643\ eV}$ difference Fourier map. To rule out small amounts of contaminating MoFe protein, an electron paramagnetic resonance (EPR) spectrum of the Fe protein used in the no MoFe protein control reaction was acquired (*Figure 2—figure supplement 3*); no signal corresponding to the $S = 3/2$ state of the FeMo-cofactor is observed. Additionally, the Fe protein used in the control was subjected to acetylene turnover conditions with no added MoFe protein. No ethylene formation was detected by gas chromatography, consistent with the absence of the MoFe protein. Performing the no MoFe protein reaction at lower KSeCN concentrations (11 and 1 mM KSeCN) resulted in a significant decrease in the intensities of the anomalous signals corresponding to the chalcogenide positions in the higher energy (12,668 eV) anomalous difference Fourier map, reflecting less Se-incorporation at the cluster (*Figure 2e, f* and *Table 1*, entries 3 and 4). Having established that the MoFe protein is not required for Se-incorporation at the Fe protein, the nucleotide dependence of the reaction was examined. Omitting both the MoFe protein and ATP regeneration system from the reaction did not yield crystals suitable for XRD studies. To obtain suitable crystals for XRD, the control reaction was repeated,

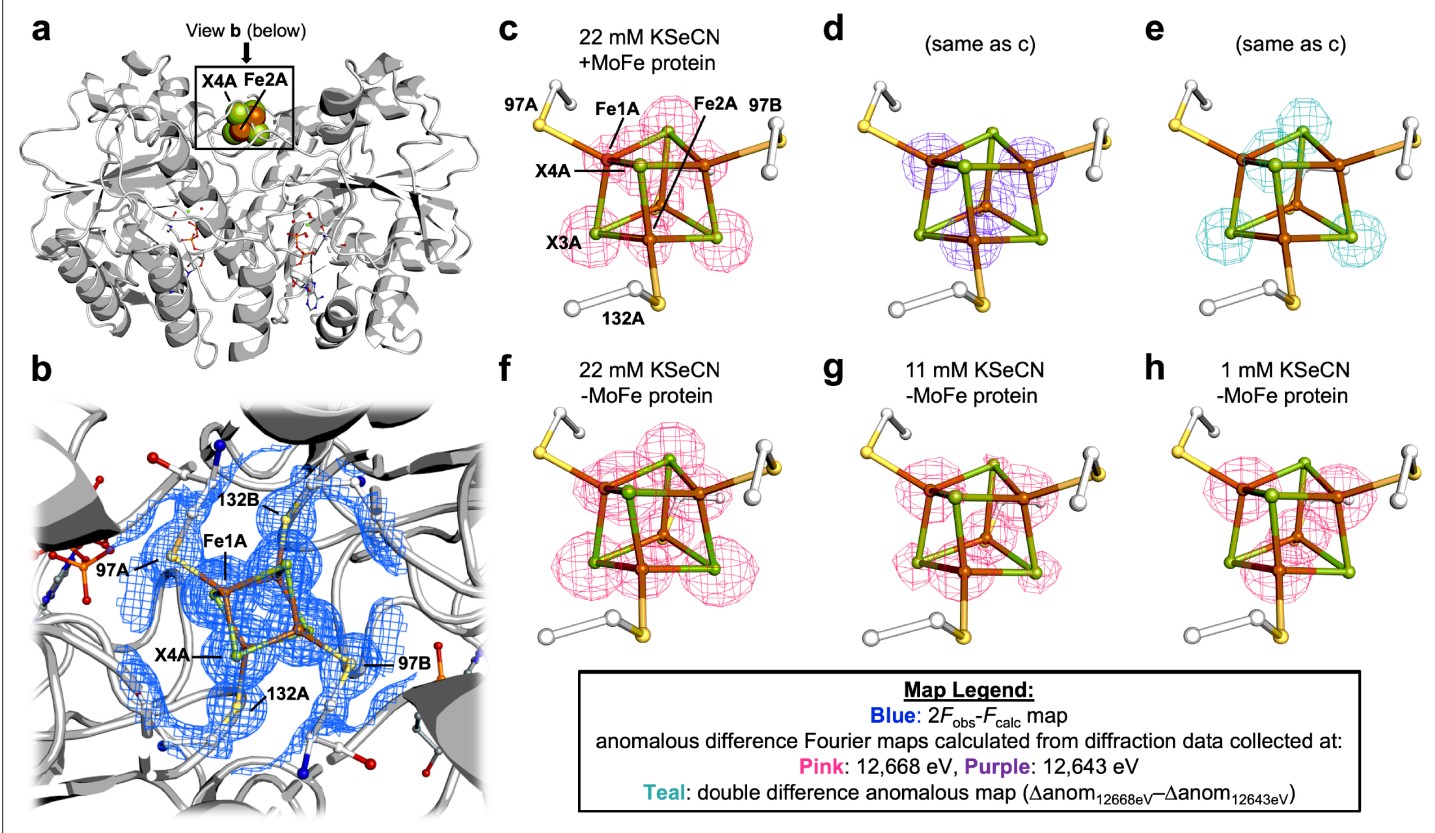

**Figure 2.** Pymol representation of the Se-incorporated Fe protein cluster at 1.51 Å resolution (PDB ID 7T4H), where the cluster chalcogenide (X) positions (green) feature a mixture of S and Se ions. (**a**) Protein overview (**b**) with overlaid electron density ($2F_{obs} - F_{calc}$) map around the $Fe_4S_4$ cluster contoured at 1.5 σ (blue mesh) viewed with the dimer twofold axis coincident with, and perpendicular to the plane of the paper, respectively. Anomalous difference Fourier maps calculated from diffraction data collected at (**c**) 12,668 eV contoured at 11.0 σ (magenta mesh, (**d**) 12,643 eV contoured at 11.0 σ (purple mesh), and (**e**) double difference ($\Delta anom_{12,668\,eV} - \Delta anom_{12,643\,eV}$)) anomalous map contoured at 11.0 σ (teal mesh). (**f–h**) Anomalous difference Fourier maps calculated from diffraction data collected at 12,668 eV (magenta mesh) corresponding to crystals derived from reactions containing 22 mM KSeCN (PDB ID 7TNE), 11 mM KSeCN (PDB ID 7TPN), and 1 mM KSeCN (PDB ID 7TPO) contoured at 11.0, 7.0, and 5.0 σ, respectively.

The online version of this article includes the following figure supplement(s) for figure 2:

**Figure supplement 1.** Comparison of $2F_{obs} - F_{calc}$ and $F_{obs} - F_{calc}$ maps for cluster modeled as exclusively S- vs. Se-containing forms.

**Figure supplement 2.** Anomalous difference Fourier maps (pink mesh, contoured at 7 σ) calculated from diffraction data collected at 12,668 eV for Se-free Fe protein crystals corresponding to crystal in (**b**) *Supplementary file 2*, PDB ID 7TPW (**c**) *Supplementary file 2*, PDB ID 7TPX, (**d**) *Supplementary file 2*, PDB ID 7TPY (nucleotide free reaction), and (**e**) *Supplementary file 2*, PDB ID 7TPZ (MgADP in place of MgATP/ATP regeneration system).

**Figure supplement 3.** Electron paramagnetic resonance (EPR) spectrum of Fe protein used in control reaction with no MoFe protein with (**A**) experimental data (black) and (**B**) simulation (orange).

**Figure supplement 4.** Anomalous difference Fourier map (pink mesh, contoured at 4.0 σ) calculated from diffraction data collected at 12,668 eV for ADP-bound Fe protein crystal soaked with KSeCN over (**a**) the entire protein, (**b**) the cluster.

**Figure supplement 5.** The FeMo-cofactor of Se-incorporated MoFe protein overlaid with the double difference ($\Delta anom_{12668eV} - \Delta anom_{12643eV}$) anomalous difference Fourier map contoured at 9.0 σ (olive mesh); X denotes a mixture of Se and S atoms.

**Figure supplement 6.** Fluorescence scan collected around Se K-edge energy for Se-incorporated Fe protein crystal.

**Figure supplement 7.** Electron paramagnetic resonance (EPR) spectrum of purified Se-incorporated Fe protein with (**A**) experimental data (black) and (**B**) simulation (orange).

followed by addition of MgADP during the reaction workup to form the MgADP-bound state for crystallization. No Se-incorporation is observed in the anomalous difference Fourier map calculated from data collected at 12,668 eV, when the Fe protein and KSeCN are the sole components of the reaction (see *Supplementary file 2*, PDB ID 7TPY and *Figure 2—figure supplement 2d*). Additionally, when MgADP and KSeCN, but no MoFe protein or ATP regeneration system, are mixed with the Fe protein, no Se-incorporation at the $Fe_4S_4$ cluster occurs (*Supplementary file 2*, PDB ID 7TPZ and

**Table 1.** Summary of crystallographically determined Se occupancies for KSeCN-derived Se-incorporation at the Fe protein cluster under various conditions.

The occupancies for the X3 and X4 chalcogenide positions were determined in triplicate[†] by analyzing three crystals prepared from a specified set of reaction conditions. For occupancy values corresponding to individual crystals, please see *Supplementary file 3*.

| Entry | Brief description of reaction conditions* | X3 occupancy (average + standard deviation) | X4 occupancy (average + standard deviation) |
|---|---|---|---|
| 1 | 22 mM KSeCN, w/ MoFe protein | 0.51 ± 0.09 | 0.43 ± 0.06 |
| 2 | 22 mM KSeCN | 0.58 ± 0.03 | 0.38 ± 0.05 |
| 3 | 11 mM KSeCN | 0.07 ± 0.02 | 0.06 ± 0.03 |
| 4 | 1 mM KSeCN | 0.02 ± 0.01 | 0.02 ± 0.01 |

*See Methods for full description.
[†]With the exception of entry 2 for which four crystals were analyzed.

*Figure 2—figure supplement 2e*). Finally, in an attempt to observe a potential ligand-bound form of the $Fe_4S_4$ cluster, the MgADP-bound crystal form was soaked with KSeCN; no density corresponding to $^-$SeCN, either near the $Fe_4S_4$ cluster or anywhere else in the protein structure, was observed (*Figure 2—figure supplement 4*).

## Discussion

The ability of iron–sulfur cluster containing metalloproteins to undergo a variety of cluster conversions and exchange reactions involving exogenous iron and sulfur species has been recognized since the pioneering work of Beinert (*Kent et al., 1982*; *Kennedy et al., 1983*; *Kennedy et al., 1984*; *Holm and Lo, 2016*). An orthogonal method for monitoring S-exchange in clusters uses selenium as a structural surrogate of sulfur (*Reynolds and Holm, 1981*; *Moulis and Meyer, 1982*). Our group's previously reported Se-incorporation results coupled with the results described herein highlight both the utility of this approach with nitrogenase and the selectivity of this process, under KSeCN turnover conditions. While the Fe protein cluster and the two-coordinate sulfides of the FeMo-cofactor undergo Se-incorporation, the P-cluster, which has been reported to undergo redox-dependent structural changes (*Peters et al., 1997*; *Keable et al., 2018*), has not yet been observed to undergo exchange of any of the constituent sulfides.

In line with the proposal that MgATP-binding results in a conformational change that renders the cluster more accessible to ligand binding relative to the nucleotide-free or MgADP-bound states (*Lindahl et al., 1987*), Se-incorporation at the $Fe_4S_4$ cluster is only observed in the presence of MgATP. The accessibility of the Fe protein cluster (*Georgiadis et al., 1992*; *Meyer, 2008*; *Einsle and Rees, 2020*) contrasts with most $Fe_4S_4$-containing proteins that feature buried clusters, with only a few exceptions (*Georgiadis et al., 1992*; *Locher et al., 2001*). It should be noted that although the Fe protein cluster remains relatively exposed in the absence of nucleotide or in the presence of MgADP (*Figure 2a, b*), incubation with KSeCN does not result in S/Se-exchange under these conditions (*Figure 2—figure supplement 2d, e*). Consequently, the position of the cluster near the surface of the protein is not a sufficient condition for KSeCN-derived Se-incorporation. These observations highlight the MgATP-dependent nature of the Fe protein as a means of regulating the physiological properties of the cluster and cluster atom exchange.

While the crystallographic observations described herein unambiguously establish the occurrence of chalcogenide exchange at the Fe protein cluster, the mechanism of this reaction remains open. The ability of Fe protein to reduce $CO_2$-to-CO (*Rebelein et al., 2017*), in the absence of the MoFe protein, suggests that the $Fe_4S_4$ cluster may coordinate $CO_2$ (*Rettberg et al., 2019*). Furthermore, the first observed instance of $N_2$ bound to a synthetic FeS cluster (a $MoFe_3S_4$ cubane) was recently reported (*McSkimming and Suess, 2021*), demonstrating that relatively simple FeS clusters can coordinate exogenous ligands (*Brown and Suess, 2022*). In the context of MoFe protein-independent $CO_2$ reduction and ligand binding to synthetic clusters, KSeCN can be viewed as a substrate analog to $CO_2$, with the Se-exchange mechanism proceeding by initial $^-$SeCN binding to an Fe center, followed

by Se–C bond cleavage, and chalcogenide exchange. Finally, while we have not probed the catalytic properties of the (partially) Se-incorporated Fe protein, Ribbe et al. recently described the redox and catalytic properties of a fully $Fe_4Se_4$-reconstituted Fe protein (*Solomon et al., 2022*). In short, the $Fe_4Se_4$-reconstituted Fe protein exhibited poorer catalytic activity relative to the native protein (*Solomon et al., 2022*), which is consistent with the poor KSeCN reduction activity previously reported by our group given the likelihood that Se-incorporated Fe protein was also being generated under these conditions (*Spatzal et al., 2015*). As highlighted in this work, any future models of substrate reduction by nitrogenase should consider the possibility that the Fe protein cluster is noninnocent with respect to substrate binding.

# Materials and methods

## Key resources table

| Reagent type (species) or resource | Designation | Source or reference | Identifiers | Additional information |
|---|---|---|---|---|
| Strain, strain background (*Azotobacter vinelandii*, Lipman) | OP | ATCC | 13705 | |

## General considerations

All protein manipulations were carried out using standard Schlenk or anaerobic tent techniques under an atmosphere of Ar or 97/3% $Ar/H_2$ mixture, respectively. Potassium selenocyanate (KSeCN) was purchased from Sigma-Aldrich. All other reagents were purchased from commercial vendors and used without further purification unless otherwise stated.

## Growth of *Azotobacter vinelandii* and nitrogenase purification

*A. vinelandii* Lipman (ATCC 13705, strain designation OP) growth and nitrogenase purification were performed based on previously published methods (*Spatzal et al., 2011*; *Spatzal et al., 2014*) with the following modifications. All protein buffers (pH 7.8) were deoxygenated, kept under an argon atmosphere, and contained 5 mM dithionite ($Na_2S_2O_4$). The supernatant from the centrifuged cell lysate was loaded onto a Q Sepharose fast flow column (GE Healthcare). In vitro nitrogenase activity was determined by monitoring acetylene reduction to ethylene as previously described (*Spatzal et al., 2015*). Ethylene and acetylene were quantified using gas chromatography (activated alumina 60/90 mesh column, flame ionization detector). MoFe protein had a specific activity of 2940 ± 30 nmol min$^{-1}$ mg$^{-1}$ ($V_{max}$) and Fe protein had a specific activity of 1880 ± 90 nmol min$^{-1}$ mg$^{-1}$ ($V_{max}$) when measured by acetylene reduction at saturation of each component.

## Preparation of Se-incorporated nitrogenase proteins using KSeCN

The Se-incorporated proteins were prepared using a previously reported protocol (*Spatzal et al., 2015*), with the following modifications. To generate sufficient material for EPR spectroscopy or crystallization, two parallel 12 ml reactions (each containing 1.5 mg of MoFe protein and 1.65 mg of Fe protein [component ratio of 2]) were combined and concentrated under argon overpressure using an Amicon filtration cell with a molecular weight cutoff of 100 kDa. The resultant concentrated protein was used to crystallize Se-incorporated MoFe protein. The corresponding 100 kDa *filtrate* was collected, and resubjected to concentration under argon overpressure using an Amicon filtration cell with a molecular weight cutoff of 30 kDa. The latter batch of concentrated protein was used to crystallize Se-incorporated Fe protein. Note that the filter membranes did not completely separate the Se-incorporated proteins (as determined by sodium dodecyl sulfate–polyacrylamide gel electrophoresis; regardless, selective crystallization of either protein was successful (*vide infra*)).

## Control KSeCN reactions with no MoFe protein

The procedure for the various control reactions was identical to that of the preparation of Se-incorporated nitrogenase proteins described above with the following changes noted. No MoFe protein was included in the control reactions. Because the MoFe protein was absent in these reactions, a 30-kDa filter membrane was used to concentrate the reaction mixture for crystallization. In addition, for the no-nucleotide control, the components of the ATP regeneration system were excluded and the resultant concentrated protein was rinsed with a 5-mM MgADP solution (3 × 8 ml) for crystallization

purposes. Finally, for the MgADP control, the ATP regeneration system was replaced with a 5-mM MgADP solution.

## Crystallization and data collection of Se-incorporated MoFe protein

The Se-incorporated MoFe protein was crystallized by the sitting-drop vapor diffusion method at ambient temperature in an inert gas chamber. The reservoir solution contained 15–20% polyethylene glycol (PEG) 4000, 0.5–0.8 M NaCl, 0.2 M imidazole/malate (pH 8.0), and 5 mM dithionite. Additionally, native MoFe protein crystals (crushed using a seed bead Eppendorf tool with either a plastic bead or glass beads) were used as seeds to accelerate the crystallization process and improve the overall crystal quality. For flash-cooling, 2-methyl-2,4-pentanediol (MPD) was either added directly to the crystal droplet, yielding 10% MPD, or the crystals were transferred into a harvesting solution consisting of the reservoir solution and 10% MPD. Complete sets of diffraction data were collected at the Synchrotron Radiation Lightsource (SSRL) beamline 12-2 equipped with a Dectris Pilatus 6 M detector. Two sets of anomalous diffraction data were collected above and below the Se K-edge at 12,668 eV (0.978690 Å) and 12,643 eV (0.980620 Å), respectively. Data were indexed, integrated, and scaled using iMosflm, XDS, and Aimless (*Leslie, 2006*; *Kabsch, 2010*; *Evans, 2006*). Phase information were obtained using the available 1.00 Å resolution structure (PDB: 3U7Q) as a molecular replacement model, omitting the metalloclusters and water from 3U7Q. Structural refinement, and rebuilding were accomplished by using REFMAC5/PHENIX, and COOT, respectively (*Murshudov et al., 1997*; *Emsley et al., 2010*; *Liebschner et al., 2019*). Neutral atomic scattering factors were used in the refinement. Anomalous difference Fourier maps were calculated using CAD/FFT in the CCP4 suite. The double difference anomalous Fourier maps were calculated using SFTOOLS (CCP4). Protein structures were displayed in PYMOL.

Consistent with our previously published MoFe protein structures containing Se-incorporated FeMo-cofactor (*Spatzal et al., 2015*; *Henthorn et al., 2019*), this structure revealed that (1) the belt sulfides were labile, with Se-incorporation predominantly at the 2B site, but also at the 5A and 3A sites (*Figure 2—figure supplement 5*) and (2) no Se-incorporation occurs at the P-cluster.

## Preparation, crystallization, and data collection of Se-incorporated Fe protein

Se-incorporated Fe protein was crystallized by the sitting-drop vapor diffusion method at ambient temperature in an inert gas chamber. The reservoir solution contained 36–41% PEG 400, 0.1–0.3 M NaCl, 0.1 M 4-(2-hydroxyethyl)-1-piperazineethanesulfonic acid (pH 7.5), 2.5 mM dithionite, and 0.17 mM 7-cyclohexyl-1-heptyl-β-D-maltoside (Cymal 7). The same parameters for data collection and refinement as Se-incorporated MoFe protein were used, with the following modifications: phase information was obtained using PDB coordinate set 6N4L as the Fe protein molecular replacement model, with the cluster, MgADP, and water molecules omitted. Cluster modeling was accomplished by modeling individual X (X = Se, S) and Fe ions at the respective cluster positions and by inputting bond distance and bond angle restraints, based on the core cluster metrics determined for synthetic clusters (SIMNOR10 and COZXUK), into the PHENIX.REFINE configuration (*Hagen et al., 1984*; *Yu et al., 1991*). The $f' = -6.00$ and $f'' = 4.00$ values for Se were used, with the latter value matching well with the fluorescence scans of Se-incorporated Fe protein crystals (see *Figure 2—figure supplement 6* for sample fluorescence scan). Se occupancies were determined by fixing the cluster atom *B*-factors to the value the Fe atoms refined to during an initial refinement. Given that *B*-factors and occupancies are correlated and the fact that there is minimal difference between the S and Fe cluster atom *B*-factors in Se-free crystals (see *Figure 2—figure supplement 2* and *Supplementary file 2*), this approach is reasonable. Neutral atomic scattering factors were used in the refinement. Anomalous difference Fourier maps were calculated using CAD/FFT in the CCP4 suite. The double difference anomalous Fourier maps were calculated using SFTOOLS (CCP4). Protein structures were displayed in PYMOL. Given restrictions regarding cluster notation as determined by the PDB, the individual atom notation in our models was converted to the cluster (SFS or SF4) format for the purposes of depositing the structures into the PDB. While the two-cluster model accurately reflects the occupancies at the distinct chalcogenide sites (X3 and X4) determined upon refinement with the individual atom cluster notation, we recognize that the two-cluster model does not realistically reflect the data and that a mixture

of partially occupied Se-incorporated clusters is likely, that is $Fe_4S_4$, $Fe_4S_3Se$, $Fe_4S_2Se_2$, $Fe_4SSe_3$, and $Fe_4Se_4$ may all be present to yield the crystallographically determined occupancies.

The structural models and structure factors have been deposited with the Protein Data Bank (PDB) under accession codes 7TPW, 7TPX, 7TPY, 7TPZ, 7T4H, 7TQ0, 7TQ9, 7TQC, 7TNE, 7TQE, 7TQF, 7TPN, 7TQH, 7TQI, 7TPO, 7TQJ, 7TQK, and 7TPV. For tables with data collection and refinement statistics, please see *Supplementary files 4–9*.

### KSeCN-soaking of Fe protein crystals

The MgADP-bound crystal form of the Fe protein was soaked with KSeCN (5 mM) by adding KSeCN directly to a crystal well, resealing, and allowing the well to sit for various lengths of time. The particular dataset provided here was obtained after the crystals had been soaked with KSeCN for 1 week.

### Purified Se-incorporated Fe protein EPR sample preparation

Se-labeled protein from three KSeCN reaction sets were combined and loaded onto an anaerobic 1 ml HiTrap Q anion exchange column (previously equilibrated with 50 mM Tris/HCl buffer [pH = 7.8] which contained 150 mM NaCl [low salt] and 5 mM dithionite). Se-incorporated MoFe protein and Se-incorporated Fe protein eluted with a linear NaCl gradient at 280 and 430 mM NaCl, respectively. Se-incorporated Fe protein was concentrated to approximately 16 mg/ml under argon overpressure using an Amicon filtration cell with a molecular weight cutoff of 30 kDa. The EPR sample was prepared as an approximately 50 µM frozen glass of Se-incorporated Fe protein in a 50:50 mixture of buffer:ethylene glycol. The buffer solution consisted of 200 mM NaCl and 50 mM Tris/HCl (pH = 7.8) and contained 25 mM dithionite (7.5 mM dithionite in EPR sample overall).

### CW EPR spectroscopy

X-band EPR spectra were obtained on a Bruker EMX spectrometer equipped with an ER 4116 DM Dual Mode resonator operated in perpendicular mode at 10 K using an Oxford Instruments ESR900 helium flow cryostat. Bruker Win-EPR software (ver. 3.0) was used for data acquisition. Spectra were simulated using the EasySpin (*Stoll and Schweiger, 2006*) simulation toolbox (release 5.2.28) with Matlab 2020b.

### Discussion of EPR data

The EPR spectrum of the Fe protein features an $S = 1/2$ signal corresponding to the $[Fe_4S_4]^{1+}$ state of the cluster with $g = [2.05, 1.94, 1.88]$ (*Lindahl et al., 1987*). While the Fe protein can exist in the $S = 3/2$ and $S = 1/2$ states, the population of the spin state depends on the sample conditions, including the presence of nucleotide and solvent. In 50% ethylene glycol, used as a cryoprotectant, most of the Fe protein cluster is in the $S = 1/2$ state (*Lindahl et al., 1985*).

The mixture of S/Se-labeled Fe protein could be separated from the MoFe protein using anion exchange chromatography and subjected to EPR spectroscopy. Based on the crystallographic data, we anticipate that the Se-labeled Fe protein exists in a mixture of Se-containing cluster states (i.e., $Fe_4S_4$, $Fe_4S_3Se$, $Fe_4S_2Se_2$, $Fe_4SSe_3$, and $Fe_4Se_4$ may all be present). As such, a familiar $g = 2$ signal corresponding to the $[Fe_4S_4]^{1+}$ cluster of the Fe protein was observed (*Figure 2—figure supplement 7*). While there are slight differences in the EPR spectra between the all-S vs. $Fe_4X_4$ (X = S, Se) mixture of the Fe protein cluster, the signal of the -S/-Se mixture could be successfully simulated using the same parameters as the all-S-containing Fe protein cluster (*Buscagan et al., 2021*). One plausible interpretation of our EPR data is that the various $Fe_4X_4$ states yield nearly identical, overlapping, signals consistent with the observation that EPR spectra of $Fe_4S_4$ vs. $Fe_4Se_4$ clusters are nearly identical (*Bobrik et al., 1978*). Alternatively, it has been recently reported that an Fe protein with an $Fe_4Se_4$ cluster is reduced to the all ferrous state in the presence of dithionite, rendering it EPR silent in perpendicular mode EPR (*Solomon et al., 2022*). In this context, the signal observed in *Figure 2—figure supplement 7* may correspond to the $[Fe_4S_4]^{1+}$ state while the $[Fe_4Se_4]^0$ state is not observed. Our results cannot distinguish between these two possible interpretations.

## Acknowledgements

We dedicate this paper to Prof James B Howard and are grateful for our extensive exchanges on cluster atom exchanges—Jim shaped the way we think about the Fe protein and our journeys with

nitrogenase. We also thank Dr Javier Fajardo Jr, Dr Shabman Hematian, and team nitro (Dr Stephanie Threatt, Dr Siobhán MacArdle, Dr Rebeccah Warmack and Ailiena Maggiolo) for insightful discussions, Jeffrey Lai for growing Azotobacter vinelandii, and Dr Paul Oyala for EPR training and support. The authors are grateful to the Gordon and Betty Moore Foundation, Don and Judy Voet, and the Beckman Institute at Caltech for their generous support of the Molecular Observatory at Caltech. Use of the Stanford Synchrotron Radiation Lightsource, SLAC National Accelerator Laboratory, is supported by the U.S. Department of Energy, Office of Science, Office of Basic Energy Sciences under Contract No. DE-AC02-76SF00515. The SSRL Structural Molecular Biology Program is supported by the DOE Office of Biological and Environmental Research, and by the National Institutes of Health, National Institute of General Medical Sciences (including P41GM103393). The Caltech EPR Facility is supported by NSF-1531940. This work was supported by the National Institutes of Health (NIH Grant GM45162) and the Howard Hughes Medical Institute (HHMI). This article is subject to HHMI's Open Access to Publications policy. HHMI lab heads have previously granted a nonexclusive CC BY 4.0 license to the public and a sublicensable license to HHMI in their research articles. Pursuant to those licenses, the author-accepted manuscript of this article can be made freely available under a CC BY 4.0 license immediately upon publication.

## Additional information

### Funding

| Funder | Grant reference number | Author |
|---|---|---|
| Howard Hughes Medical Institute | | Douglas C Rees |
| National Institutes of Health | GM45162 | Douglas C Rees |

The funders had no role in study design, data collection, and interpretation, or the decision to submit the work for publication.

### Author contributions

Trixia M Buscagan, Conceptualization, Data curation, Formal analysis, Validation, Investigation, Visualization, Methodology, Writing – original draft, Writing – review and editing; Jens T Kaiser, Data curation, Validation, Methodology, Writing – review and editing; Douglas C Rees, Conceptualization, Supervision, Funding acquisition, Methodology, Project administration, Writing – review and editing

### Author ORCIDs

Trixia M Buscagan ⓘ http://orcid.org/0000-0001-8242-9203
Jens T Kaiser ⓘ http://orcid.org/0000-0002-5948-5212
Douglas C Rees ⓘ http://orcid.org/0000-0003-4073-1185

### Decision letter and Author response

Decision letter https://doi.org/10.7554/eLife.79311.sa1
Author response https://doi.org/10.7554/eLife.79311.sa2

## Additional files

### Supplementary files

• MDAR checklist

• Supplementary file 1. Comparison of B-factors for cluster modeled as exclusively FeS vs FeSe forms.

• Supplementary file 2. B-factor analysis of cluster atoms in Se-free Fe protein crystal structures.

• Supplementary file 3. Summary of crystallographically determined Se occupancies for KSeCN derived Se-incorporation at the Fe protein cluster under various conditions.

• Supplementary file 4. Data collection and refinement statistics for *Se-free* Fe protein crystals from

various control reactions.

• Supplementary file 5. Data collection and refinement statistics for Se-incorporated Fe protein crystals derived from *22 mM* KSeCN reaction in the *presence* of MoFe protein.

• Supplementary file 6. Data collection and refinement statistics for Se-incorporated Fe protein crystals derived from *22 mM* KSeCN reaction in the *absence* of MoFe protein.

• Supplementary file 7. Data collection and refinement statistics for Se-incorporated Fe protein crystals derived from *11 mM* KSeCN reaction in the *absence* of MoFe protein.

• Supplementary file 8. Data collection and refinement statistics for Se-incorporated Fe protein crystals derived from *1 mM* KSeCN reaction in the *absence* of MoFe protein.

• Supplementary file 9. Data collection and refinement statistics for Fe protein crystal soaked with KSeCN.

## Data availability

Diffraction data have been deposited in the RCSB PDB under the accession codes 7TPW, 7TPX, 7TPY, 7TPZ, 7T4H, 7TQ0, 7TQ9, 7TQC, 7TNE, 7TQE, 7TQF, 7TPN, 7TQH, 7TQI, 7TPO, 7TQJ, 7TQK, and 7TPV.

The following datasets were generated:

| Author(s) | Year | Dataset title | Dataset URL | Database and Identifier |
|---|---|---|---|---|
| Buscagan, Kaiser Rees | 2022 | SELENIUM INCORPORATED NITROGENASE FE-PROTEIN (AV2-SE) FROM A. VINELANDII | https://www.rcsb.org/ 7T4H | RCSB Protein Data Bank, 7T4H |
| Buscagan, Kaiser Rees | 2022 | SELENIUM-INCORPORATED NITROGENASE FE-PROTEIN (AV2-SE) FROM A. VINELANDII | https://www.rcsb.org/ 7TNE | RCSB Protein Data Bank, 7TNE |
| Buscagan, Kaiser Rees | 2022 | SELENIUM-INCORPORATED NITROGENASE FE PROTEIN (AV2-SE) FROM A. VINELANDII (11 MM KSECN) | https://www.rcsb.org/ 7TPN | RCSB Protein Data Bank, 7TPN |
| Buscagan, Kaiser Rees | 2022 | SELENIUM-INCORPORATED NITROGENASE FE-PROTEIN (AV2-SE) FROM A. VINELANDII (1MM KSECN) | https://www.rcsb.org/ 7TPO | RCSB Protein Data Bank, 7TPO |
| Buscagan, Kaiser Rees | 2022 | SELENIUM-FREE NITROGENASE FE PROTEIN (AV2) FROM A. VINELANDII (5MM KSECN SOAKED) | https://www.rcsb.org/ 7TPV | RCSB Protein Data Bank, 7TPV |
| Buscagan, Kaiser Rees | 2022 | SELENIUM-FREE NITROGENASE FE PROTEIN (AV2) FROM A. VINELANDII | https://www.rcsb.org/ 7TPW | RCSB Protein Data Bank, 7TPW |
| Buscagan, Kaiser Rees | 2022 | SELENIUM-FREE NITROGENASE FE PROTEIN (AV2) FROM A. VINELANDII | https://www.rcsb.org/ 7TPX | RCSB Protein Data Bank, 7TPX |

*Continued on next page*

*Continued*

| Author(s) | Year | Dataset title | Dataset URL | Database and Identifier |
|---|---|---|---|---|
| Buscagan, Kaiser Rees | 2022 | SELENIUM-FREE NITROGENASE FE PROTEIN (AV2) FROM A. VINELANDII (NUCLEOTIDE CONTROL) | https://www.rcsb.org/7TPY | RCSB Protein Data Bank, 7TPY |
| Buscagan, Kaiser Rees | 2022 | SELENIUM-FREE NITROGENASE FE PROTEIN (AV2) FROM A. VINELANDII (NUCLEOTIDE CONTROL) | https://www.rcsb.org/7TPZ | RCSB Protein Data Bank, 7TPZ |
| Buscagan, Kaiser Rees | 2022 | SELENIUM-INCORPORATED NITROGENASE FE PROTEIN (AV2-SE) FROM A. VINELANDII (22 MM KSECN, WITH AV1) | https://www.rcsb.org/7TQ0 | RCSB Protein Data Bank, 7TQ0 |
| Buscagan, Kaiser Rees | 2022 | SELENIUM-INCORPORATED NITROGENASE FE PROTEIN (AV2-SE) FROM A. VINELANDII (22 MM KSECN, WITH AV1) | https://www.rcsb.org/7TQ9 | RCSB Protein Data Bank, 7TQ9 |
| Buscagan, Kaiser Rees | 2022 | SELENIUM-INCORPORATED NITROGENASE FE PROTEIN (AV2-SE) FROM A. VINELANDII (22 MM KSECN) | https://www.rcsb.org/7TQC | RCSB Protein Data Bank, 7TQC |
| Buscagan, Kaiser Rees | 2022 | SELENIUM-INCORPORATED NITROGENASE FE PROTEIN (AV2-SE) FROM A. VINELANDII (22 MM KSECN) | https://www.rcsb.org/7TQE | RCSB Protein Data Bank, 7TQE |
| Buscagan, Kaiser Rees | 2022 | SELENIUM-INCORPORATED NITROGENASE FE PROTEIN (AV2-SE) FROM A. VINELANDII (22 MM KSECN) | https://www.rcsb.org/7TQF | RCSB Protein Data Bank, 7TQF |
| Buscagan, Kaiser Rees | 2022 | SELENIUM-INCORPORATED NITROGENASE FE PROTEIN (AV2-SE) FROM A. VINELANDII (11 MM KSECN) | https://www.rcsb.org/7TQH | RCSB Protein Data Bank, 7TQH |
| Buscagan, Kaiser Rees | 2022 | SELENIUM-INCORPORATED NITROGENASE FE PROTEIN (AV2-SE) FROM A. VINELANDII (11 MM KSECN) | https://www.rcsb.org/7TQI | RCSB Protein Data Bank, 7TQI |
| Buscagan, Kaiser Rees | 2022 | SELENIUM-INCORPORATED NITROGENASE FE PROTEIN (AV2-SE) FROM A. VINELANDII (1 MM KSECN) | https://www.rcsb.org/7TQJ | RCSB Protein Data Bank, 7TQJ |

*Continued*

| Author(s) | Year | Dataset title | Dataset URL | Database and Identifier |
|---|---|---|---|---|
| Buscagan, Kaiser Rees | 2022 | SELENIUM-INCORPORATED NITROGENASE FE PROTEIN (AV2-SE) FROM A. VINELANDII (1 MM KSECN) | https://www.rcsb.org/ 7TQK | RCSB Protein Data Bank, 7TQK |

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
