## [Editor Report]

This manuscript describes the unexpected observation of selenium exchange into an iron-sulfur cluster cofactor of a component of nitrogenase. The work sets the stage for future mechanistic study of this phenomenon. It also provides a roadmap for study of sulfide exchange in other classes of iron-sulfur cluster enzymes.

---

## [Decision Letter]

**Decision letter after peer review:**

Thank you for submitting your article "Selenocyanate Derived Se-Incorporation into the Nitrogenase Fe Protein Cluster" for consideration by *eLife*. Your article has been reviewed by 2 peer reviewers, including Amie Boal as the Reviewing Editor and Reviewer #1, and the evaluation has been overseen by José Faraldo-Gómez as the Senior Editor.

Essential revisions:

1) While both reviewers agree that the findings of the study are exciting and presented effectively, there are some concerns about the broader significance of the work. To address this issue, we recommend that a revision include the two simple experiments suggested by reviewer #2 – testing the Se-substituted Fe protein for support of catalysis by the MoFe protein and evaluating the reversibility of the Se incorporation. These efforts will extend the work beyond simply reporting the Se substitution phenomenon.

2) The authors should discuss the recent manuscript by Ribbe et al., as suggested by reviewer #2.

*Reviewer #1 (Recommendations for the authors):*

This is a well-written manuscript with excellent presentation of results. I am in favor of publication. My only concern is that the study might not be of sufficiently broad interest for publication in a journal such as *eLife*? Perhaps the authors could try to give some additional examples of broader applications of the work to address this potential weakness?

*Reviewer #2 (Recommendations for the authors):*

In general, I find the observations of the study to be exciting, but I also feel that the significance and the impact of the work could be increased if the authors included experiments on the redox properties of the Se-substituted iron protein and examined whether it is capable of supporting catalysis by the molybdenum-iron protein. While the former experiments require redox titrations (not necessarily routine for every laboratory), the latter should be fairly easy to carry out. The relevance of these experiments is particularly amplified in light of a recent study by Ribbe and colleagues (reference 35) which describes the insertion of a synthetic 4Fe-4Se cluster into the iron protein. I also think that this particular paper deserves a brief mention in the main text (which actually could complement the authors' work).

Another potentially interesting aspect of the work is the mechanism of S-Se exchange which the authors touch upon in the discussion. While delineating this mechanism may be beyond the scope of this study, experiments investigating whether the S-to-Se-substitution is reversible (through the addition of a sulfide source in the presence or absence of ATP) would be a useful addition.

One final comment is regarding the aesthetics of the figures. Being the centerpiece of the paper, Figure 2 could certainly be made a bit more attractive. I recommend sticking to the same font and overall format as in Figure 1 (Times New Roman is not a great choice). Also, the information in the map legend box should be presented in a different way (although I don't have a specific suggestion). Finally, the structure factor symbol F should be italicized throughout the figures and the text.

---

## [Author Response]

Essential Revisions (for the authors):1) While both reviewers agree that the findings of the study are exciting and presented effectively, there are some concerns about the broader significance of the work. To address this issue, we recommend that a revision include the two simple experiments suggested by reviewer #2 – testing the Se-substituted Fe protein for support of catalysis by the MoFe protein and evaluating the reversibility of the Se incorporation. These efforts will extend the work beyond simply reporting the Se substitution phenomenon.2) The authors should discuss the recent manuscript by Ribbe et al., as suggested by reviewer #2.

We thank the reviewers and editors, for their insightful and supportive feedback. We address the essential revisions and additional reviewer recommendations below. When appropriate, the changes to the manuscript are indicated or referenced. An accompanying word document of the manuscript including track changes is also attached.

As points 1 and 2 are connected, we will address them together.

The initial submission of our manuscript discussed the Ribbe et al., manuscript in the concluding remarks. In response to both reviewers, we have made adjustments to this discussion by introducing the Ribbe et al., paper earlier in our manuscript (introduction) and by highlighting additional results from the Ribbe et al., paper within the concluding paragraph. We hope this better highlights the paper in the manner requested by the reviewers.

The Ribbe and Hu paper details the preparation and characterization of the Fe protein prepared by reconstitution of the apo (cluster-free) protein with a synthetic Fe_4_Se_4_ cluster. This is a critical distinction from our study, which instead of a reconstitution, details the exchange reaction of Se for S during turnover of KSeCN using native Fe_4_S_4_ -containing Fe protein. Reconstitution of FeS cluster containing proteins is well precedented, including incorporation of selenated clusters. The surprising and significant aspect of our work is that this exchange reaction is a property of the Fe protein and not the complete nitrogenase system. The Fe protein is quite generally considered as the “dinitrogenase reductase”, highlighting its role in electron transfer as the source of reducing equivalents to the active site on the MoFe protein. Our work conclusively establishes that the Fe protein undergoes a reaction that results in modification of the cluster; this reactivity may be relevant not only to its role in metallocluster biosynthesis, but also more broadly in substrate reduction by nitrogenase. The modification to the Fe protein cluster also occurs in the presence of the MoFe protein in the context of KSeCN turnover.

The requested simple experiment would require one to two months of additional experiments, assuming the experiments proceed according to plan. These additional experiments would include obtaining a sufficient amount of Fe protein for crystallization. More specifically, large amounts of the native iron protein would need to be purified, transformed to the partially occupied Fe_4_Se_4_ form, the resultant Se-containing protein crystallized (which takes a couple weeks) for verification of the Se occupancy. Once the Se-containing protein is obtained and occupancy verified, the requested reaction between the Se-containing Fe protein, native MoFe protein, and acetylene (or other substrate) could be attempted, the resultant proteins (including the MoFe protein) crystallized to determine whether Se-incorporation remains consistent throughout the requested reaction. Without verification that the Se-incorporation remains consistent pre- and post-catalysis, any resultant change (or lack of change) in catalytic activity could be attributed to a change in cluster speciation. Related, we are aware of the possibility that the Se ions may migrate between the nitrogenase clusters (FeMo-cofactor and Fe_4_S_4_ cluster).

We agree that determining whether our Se-form of the Fe protein supports catalysis is valuable, but the experiment is not a simple undertaking, however (see next paragraph please). Furthermore, the Ribbe and Hu groups paper has already addressed this question; the Fe_4_Se_4_-reconstituted Fe protein exhibited poorer catalytic activity relative to the native protein, which is consistent with the poor KSeCN reduction activity previously reported by our group, given the likelihood that Seincorporated Fe protein was also being generated under these conditions.

The importance of our contribution is that it highlights the significance of studying cluster speciation as it pertains to establishing cluster atom exchange and the unique, broad, properties of the Fe protein cluster as they may pertain to the roles of the Fe protein. We have also emphasized that these results, to our knowledge, constitute the first definitive structural evidence for substrate binding at the Fe protein cluster. While a CO_2_ bound crystal form of the Fe protein isolated from *Methanosarcina acetivorans* was reported in a recent paper (added reference, currently #34 in the manuscript), the data is ambiguous, and the discussed model does not contain CO_2_ (PDB ID: 6NZJ).

Reviewer #1 (Recommendations for the authors):This is a well-written manuscript with excellent presentation of results. I am in favor of publication. My only concern is that the study might not be of sufficiently broad interest for publication in a journal such as eLife? Perhaps the authors could try to give some additional examples of broader applications of the work to address this potential weakness?

Given the current paradigm in the nitrogenase field, that substrates interact with the FeMo-cofactor active site, we believe this work constitutes a significant contribution to the bioinorganic field. In addition to reporting the first definitive structural evidence for substrate interactions at the Fe protein cluster (ATP-dependent chalcogenide exchange) we show that the MoFe protein is not required for the reaction—both of these results are unexpected as the Fe protein cluster is not typically considered to react with substrates. Here, like the two-coordinate sulfides of the FeMocofactor, we show that the three-coordinate sulfides of the Fe protein cluster exchange. Any future models of substrate reduction will need to consider the possibility that the Fe protein cluster is non-innocent with respect to substrate binding. We have added this last sentence to the concluding remarks of our paper. We have also reframed the paper/responded to the *eLife* digest questions in hopes of clarifying the broader significance of our work.

Reviewer #2 (Recommendations for the authors):In general, I find the observations of the study to be exciting, but I also feel that the significance and the impact of the work could be increased if the authors included experiments on the redox properties of the Se-substituted iron protein and examined whether it is capable of supporting catalysis by the molybdenum-iron protein. While the former experiments require redox titrations (not necessarily routine for every laboratory), the latter should be fairly easy to carry out. The relevance of these experiments is particularly amplified in light of a recent study by Ribbe and colleagues (reference 35) which describes the insertion of a synthetic 4Fe-4Se cluster into the iron protein. I also think that this particular paper deserves a brief mention in the main text (which actually could complement the authors' work).

Please see our response to essential revision 1.

Another potentially interesting aspect of the work is the mechanism of S-Se exchange which the authors touch upon in the discussion. While delineating this mechanism may be beyond the scope of this study, experiments investigating whether the S-to-Se-substitution is reversible (through the addition of a sulfide source in the presence or absence of ATP) would be a useful addition.

While we agree that studying the reversibility of the S-Se exchange would be a useful addition, we believe this study is beyond the scope of a short report in *eLife*. There are many unanswered questions with respect to the role(s) dithionite byproducts play in the speciation of the various nitrogenase clusters. In our opinion, without first establishing the S-Se exchange process in the presence of a non-S containing reductant, studying whether S-Se exchange is reversible may lead to ambiguous results i.e. the source of the S would be unclear. We speculate that using dithionite as a reductant precludes the formation of a fully occupied Fe_4_Se_4_ site.

One final comment is regarding the aesthetics of the figures. Being the centerpiece of the paper, Figure 2 could certainly be made a bit more attractive. I recommend sticking to the same font and overall format as in Figure 1 (Times New Roman is not a great choice). Also, the information in the map legend box should be presented in a different way (although I don't have a specific suggestion). Finally, the structure factor symbol F should be italicized throughout the figures and the text.

The authors thank reviewer #2 for the feedback regarding Figure 2. In response, we have changed the Figure 2 font to the same font used in Figure 1 (Arial) and made minor adjustments to improve the aesthetics of the figure although (subjectively) we find the figure attractive. We have also changed the Map Legend box.

The structure factor symbol, *F*, has been italicized throughout the figures and texts—thank you very much for catching this.